# Changing Climatic Factors Favor Dengue Transmission in Lahore, Pakistan

**Syed Ali Asad Naqvi** [1,*], **Bulbul Jan** [2], **Saima Shaikh** [3], **Syed Jamil Hasan Kazmi** [3], **Liaqat Ali Waseem** [1], **Muhammad Nasar-u-minAllah** [4] and **Nasir Abbas** [5]

1   Department of Geography, Government College University Faisalabad, Faisalabad 38000, Pakistan; drliaqataliwaseem@gcuf.edu.pk
2   The Institute of Space and Planetary Astrophysics, University of Karachi, Karachi 75270, Pakistan; bulbul.gilgit@gmail.com
3   Department of Geography, University of Karachi, Karachi 75270, Pakistan; saima-ku@uok.edu.pk (S.S.); jkazmi@usa.net (S.J.H.K.)
4   Department of Geography, Govt. Postgraduate College Gojra, Gojra 56000, Pakistan; nasarbhalli@yahoo.com
5   Department of Geography, Government College University Lahore, Lahore 54000, Pakistan; nasir_gis_geo@hotmail.com
*   Correspondence: draliasad@gcuf.edu.pk; Tel.: +92-306-674-1774

**Abstract:** Dengue fever (DF) is a national health problem in Pakistan. It has become endemic in Lahore after its recent reemergence in 2016. This study investigates the impacts of climatic factors (temperature and rainfall) on DF transmission in the district of Lahore through statistical approaches. Initially, the climatic variability was explored using a time series analysis on climatic factors from 1970 to 2012. Furthermore, ordinary and multiple linear regression analyses were used to measure the simulating effect of climatic factors on dengue incidence from 2007 to 2012. The time series analysis revealed significant annual and monthly variability in climatic factors, which shaped a dengue-supporting environment. It also showed a positive temporal relationship between climatic factors and DF. Moreover, the regression analyses revealed a substantial monthly relationship between climatic factors and dengue incidence. The ordinary linear regression of rainfall versus dengue showed monthly $R^2 = 34.2\%$, whereas temperature versus dengue presented $R^2 = 38.0\%$. The multiple regression analysis showed a monthly significance of $R^2 = 44.6\%$. Consequently, our study shows a substantial synergism between dengue and climatic factors in Lahore. The present study could help in unveiling new ways for health prediction modeling of dengue and might be applicable in other subtropical and temperate climates.

**Keywords:** time series; linear regression; rainfall; temperature; climate; DF

## 1. Introduction

Dengue fever (DF) is a viral disease that has become a major public health problem owing to a substantial rise in the incidence across the globe during the last couple of decades [1,2]. According to the World Health Organization (WHO), 2.5 billion people are at risk of its infection worldwide [3]. A newly revised assessment showed that its burden has risen up to three times relative to the previous WHO estimates [4,5]. Dengue virus (DENV) is primarily transmitted by a female mosquito *Aedes aegypti* [6]. DENV belongs to the genus *Flavivirus*, having four serotypes ranging from DEN 1 to 4, responsible for causing DF, dengue shock syndrome, and dengue hemorrhagic fever [7]. A vaccine (Dengvaxia (CYD-TDV) by Sanofi Pasteur) for dengue prevention was tested and registered in various countries, but the WHO considered it risky for seronegative individuals [8–10]. Due to the unsafe vaccine, prevention and controlling plans are the only considerable options to limit dengue.

Global climate change in recent years has increased the risks to human health [11,12]. Climate change has a strong impact on the spread of dengue and its mosquito vector [13]. Rising temperatures and varying precipitation are thought to be the major contributors to increasing dengue epidemics in various places of the world [14,15]. The distribution of dengue mosquitoes and their bite frequency and incubation period are significantly affected by the temperature. The incubation period at 27 °C is ten days, but this drops to seven days at 34 °C. This shortening of the incubation period is quite critical in terms of disease transmission [16,17].

Around 112 countries, including Pakistan, are affected by minor and major epidemics, but predominance remained in their urban and peri-urban areas [18–20]. Pakistan has faced at least seven reported epidemics during the past twenty years [20]. Dengue was first reported in Karachi during August 1994 after an unusual period of heavy rainfall [21]. It turned into an epidemic in Karachi during 2005 [22]. Its incidence increasingly hit the country afterwards by affecting 105 districts out of 146, making it a serious reportable disease [23]. From 2006 to 2011, up to 40,987 DENV infection cases and 490 deaths were reported nationwide [24]. The upper parts of Punjab Province were noticeably affected by DF, including Lahore, where it struck from 2006 to 2009, but in 2010, it ultimately became an epidemic [25].

Lahore suffered the main outbreak of dengue in 2011, when up to 11,283 cases and 257 deaths were reported [26]. Furthermore, the confirmed and suspected cases in 2016 increased significantly to 1110 and 2029, respectively, which again increased the risk of a new major dengue outbreak after 2011. These incidences of dengue were highly attributed to the climate and the urban environment. It was reported by Sajjad et al. (2009) that over the last 57 years, the mean annual temperature of Lahore has increased up to 0.89 °C [27]. Temperature index-based mapping animation revealed that Lahore district is among those regions where both dengue mosquitoes could exist [28]. The average annual rainfall of Lahore (575 mm) is quite similar to other dengue-supportive environments [29]. Consequently, Lahore district, with a fast urbanization and suitable climatic conditions, has become susceptible to dengue disease. Therefore, it is the need of the hour to measure and predict its risk, which could ultimately be beneficial for its considerable eradication.

Several studies highlight a strong relation between dengue and climate changes. For example, high temperatures, precipitation, and vegetation may influence the dengue epidemic. Many researchers used statistical analysis, i.e., linear and multiple linear regression (MLR) analysis, to examine the relationship between climatic factors and dengue incidence [30]. Lai (2018) applied MLR to study the effect of weather factors on dengue incidence over Southern Taiwan [31]. MLR models were used to observe the relations between changes in the incidence rate of dengue fever and climate variability in the warm and humid region of Mexico [32]. A correlation between climatic and environmental factors with the incidence of dengue was applied to develop prediction models over the Philippines region using remote-sensing data. The best-fitting models were then selected to describe the relationship between dengue incidence and climate variables [33]. Another study revealed the association between the monthly incidence of dengue fever and climate variables such as precipitation, temperature, humidity, and the Acre River level, using generalized autoregressive moving average (GAMA) models with a negative binomial distribution [14].

In the present study, we have tried to use statistical models to measure climate-related dengue health risks with a broad spectrum. This statistical study unveils new ways to explore dengue-related health and climatic connections in Lahore and might be applicable in other subtropical and temperate climates.

*Statement of the Problem*

In Lahore, the ever-increasing dengue burden is an environmental issue, which needs effective research and policies for its eradication. In the perspective of learning from the past and keeping in view the reemergence of dengue, we conducted the present study by utilizing the previous climate and dengue epidemic data from the years 1970–2012 and 2007–2012 to investigate the dengue risk

in relation to climatic variables. This study might be very helpful for the development of dengue risk assessment models based on statistics and can be a vital update to health risk controlling plans, especially for Lahore.

## 2. Materials and Methods

### 2.1. Study Area

The Study area of this research, "district Lahore" (Figure 1), is ranked as semi-arid climatic city district and is situated in the key group of climate category 'sub-tropical continental lowlands'. The climatic data of Lahore reflects the extreme climatic conditions in the city. Its temperature in summers is higher with rains from late monsoons [34], with temperatures of up to 48 °C in May and June. The monsoon period in Lahore is from mid-July until mid-September [35,36]. Dengue and its reemergence are critical factors, which urged us to choose Lahore as the study area.

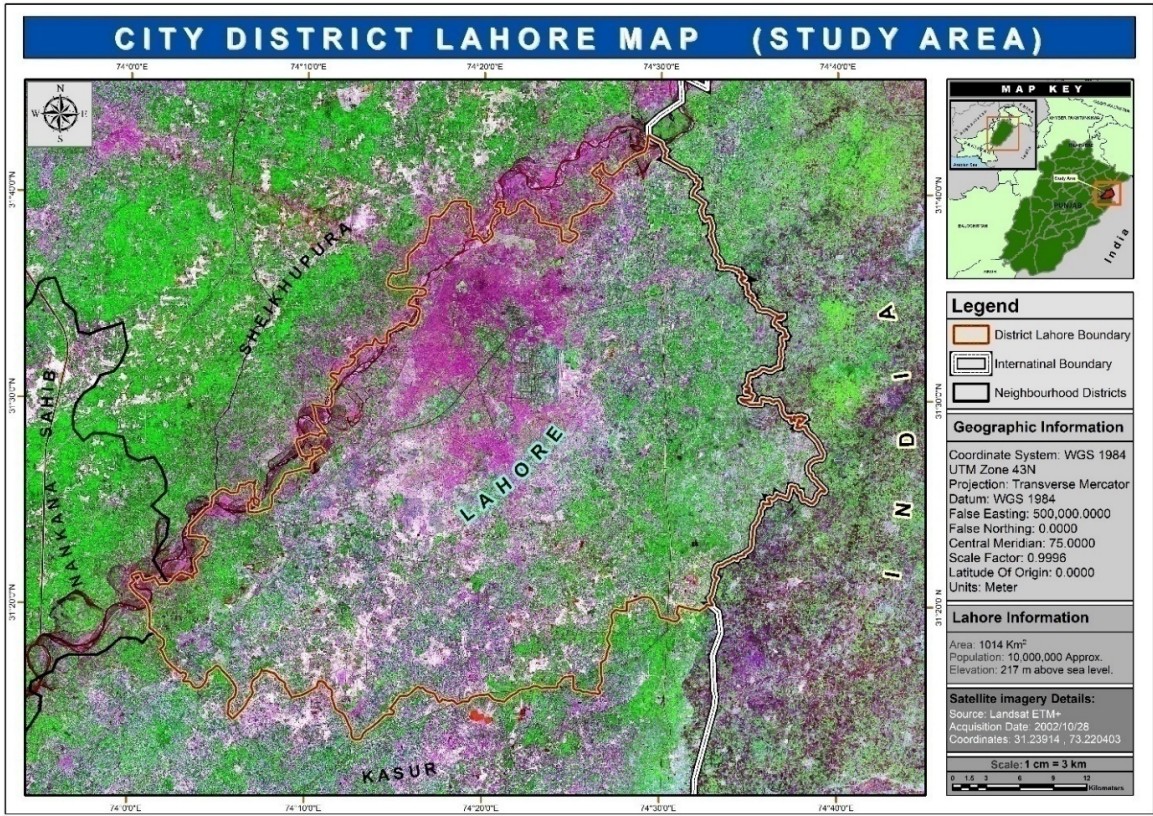

**Figure 1.** Study Area Map.

### 2.2. Data and Statistical Approaches

Two datasets were used for this study. Firstly, monthly dengue cases data during the period of 2007–2012 was obtained from the Directorate General Health Services Punjab (DGHSP) while climate data (Lahore station) of the period of 1970–2012 was collected from the Pakistan Meteorological Department (PMD). This research is divided into two sections; in the first section, climate change of Lahore is discussed, whereas in the second section, the impact of climatic factors on dengue was detected with the help of time series analysis and other statistical approaches.

The study investigated climate change and dengue–climate phenomena through time series analysis, which expresses a trend analysis [37], ordinary linear regression analysis, and multiple linear regression analysis to reveal the long-term climate change and the relationship among climatic factors and dengue. Ordinary linear regression analysis was employed to show the evaluation of the unknown

value of one variable from the known value of the other variable [38]. Mathematically, this relationship is expressed as Equation (1)

$$y = a + b \cdot x + e \tag{1}$$

where 'b' represents the slope parameter, while 'a' is the intercept parameter, and 'e' represent residuals. These parameters were found using the least square procedure Equation (2), i.e.,

$$a = \frac{\sum y_i \sum x_i^2 - \sum x_i \sum x_i y_i}{n \sum x_i^2 - (\sum x_i)^2}, b = \frac{n \sum x_i y_i - \sum x_i \sum y_i}{n \sum x_i^2 - (\sum x_i)^2} \tag{2}$$

Dengue data from 2007 to 2012 (monthly data) was used for ordinary linear and multiple linear regression analyses [39]. The multiple regression analysis was utilized to compute the impacts of temperature and rainfall on dengue. We modeled the reported cases of dengue on the basis of climatic factors, i.e., temperature and rainfall [30]. For this purpose, we considered a multiple regression analysis between the dependent variable, i.e., dengue incidence (y), and the independent variables temperature ($x_1$) and rainfall ($x_2$). This relationship is expressed as Equation (3);

$$y_{dengue} = a + b_1 \cdot x_1 + b_2 \cdot x_2 + e_j \tag{3}$$

where a, $b_1$, and $b_2$ were also estimated using the least square procedure and $e_j$ error term. To find out the variability of climatic factors (temperature and rainfall) and trend of dengue cases, time series analysis assisted to study the forecasting of future values of a time series from current and past values. It relates the actual performance and analyzes the cause of variations [39,40]. These results were verified by the coefficient of determination ($R^2$), F-test, and *p*-values [30,32].

## 3. Results

*3.1. Climate Change in Lahore (1970–2012)*

We analyzed the long-term variability of climatic factors, i.e., temperature and rainfall (1970 to 2012) of Lahore district. The results of temperature and rainfall trends over this period are as follows.

3.1.1. Temperature

During the entire 43 years (1970–2012), annual and monthly temperature trends were inclined to increase. Both annual mean maximum and minimum temperature were recorded as 25.63 °C in 2002 and 23.38 °C in 1983, respectively. The standard deviation (SD) is considered a useful technique here to show the variabilities in both temporal distributions. In the case of the annual distribution, the highest SD recorded was 8.04 in 2012, whereas the lowest was 6.66 in 1996 (Figure 2). In an overall monthly scenario, 34.27 °C and 13.45 °C were the maximum and minimum temperatures recorded in June and January, respectively. The highest SD was 1.76 in May and the lowest SD was 0.84 during August and September (Figure 3). It is evident from the plots that there is considerable monthly variation and some decadal variability from year to year.

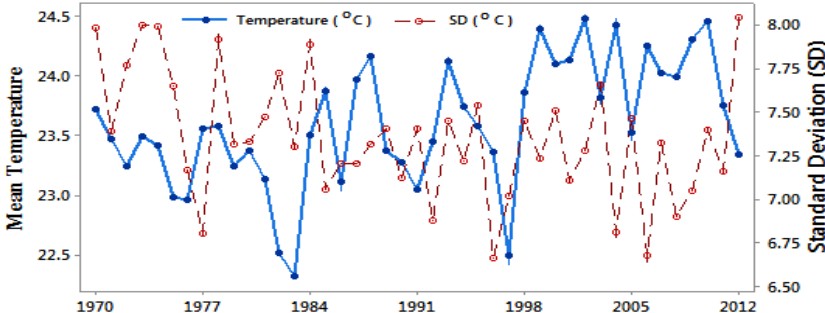

**Figure 2.** Time series of the standard deviation of the mean annual temperature of Lahore district within 1970–2012.

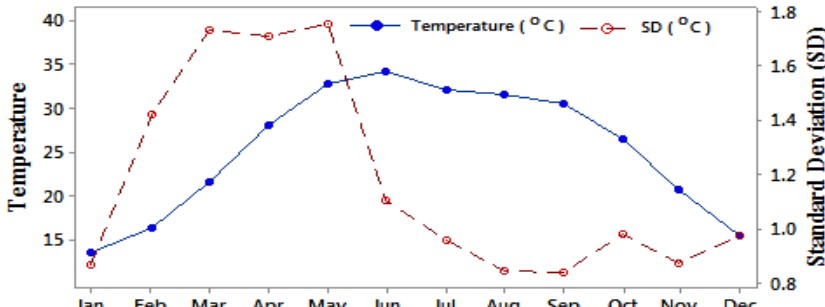

**Figure 3.** Series of the standard deviation of the mean monthly temperature of Lahore district within 1970–2012.

### 3.1.2. Rainfall

The variation in total rainfall from 1970–2012 shows a substantial change in both annual and monthly periods. The annual lowest total rainfall was 333.7 mm in 2002, whereas the highest was 1232.5 mm in 1997. In a temporal plot of SD, the highest SD of annual total rainfall was recorded as 184.96 in 1996 (Figure 4). In the monthly distribution, the total rainfall recorded was 8441.80 mm. In this context, July had the highest overall total, whereas November had the lowest and August had the highest SD (126.32) (Figure 5). Both temporal plots (annual and monthly) show a significant change from year to year and month to month. These findings suggest that climatic conditions of Lahore have been changing over the course of time.

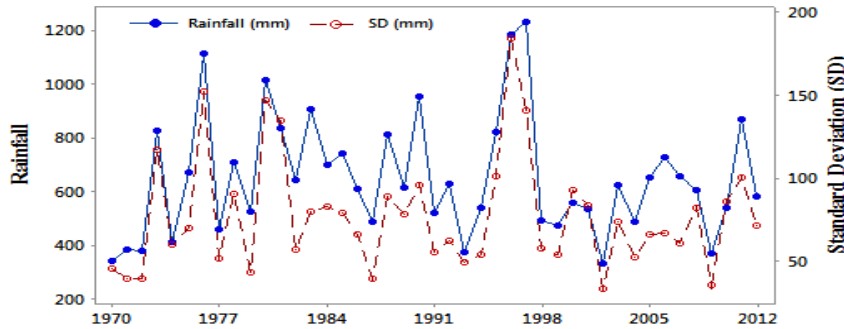

**Figure 4.** Time series of the standard deviation of the total annual rainfall of Lahore district within 1970–2012.

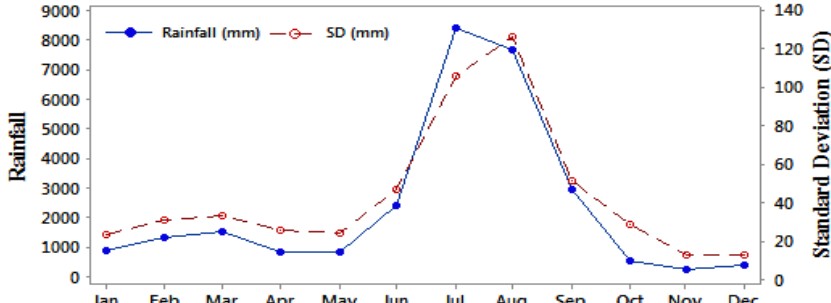

**Figure 5.** Time series of the standard deviation of total rainfall of Lahore district within 1970–2012.

### 3.2. Climatic Factors vs. DF; Time Series Analysis

The monthly average temperature, rainfall, and dengue cases recorded at study area (2007–2012) were 25.07 °C, 50.49 mm, and 239, respectively. The highest and lowest temperature recorded were 35.30 °C and 11.20 °C, respectively (Table 1). The maximum total rainfall recorded was 1052.5 mm in August, and the minimum was 11.5 mm in November, while the maximum and minimum number

of dengue cases were 6433 and 4, recorded in the months of September and February, respectively (Table 2).

**Table 1.** Descriptive Statistics of Temperature, Rainfall, and Dengue cases of Lahore from 2007–2012.

| Variables | N | Mean | SE Mean | SD | Min | Median | Max |
|---|---|---|---|---|---|---|---|
| Temperature | 72 | 25.07 | 0.834 | 7.074 | 11.20 | 27.00 | 35.300 |
| Rainfall | 72 | 50.49 | 8.47 | 74.12 | 0.00 | 18.85 | 288 |
| Dengue | 72 | 239 | 106 | 901 | 0.00 | 3.00 | 6314 |

**Table 2.** Estimation of monthly average temperature (Avg. Temp.), total rainfall, and total dengue cases of Lahore from 2007–2012.

| Variables | Jan | Feb | Mar | Apr | May | Jun | July | Aug | Sep | Oct | Nov | Dec |
|---|---|---|---|---|---|---|---|---|---|---|---|---|
| Avg. Temp. | 11.2 | 13.94 | 19.47 | 24.31 | 28.07 | 28.6 | 27.1 | 26.30 | 25.2 | 22.7 | 17.87 | 13.0 |
| Total Rainfall | 57.8 | 183.2 | 139.9 | 126.4 | 75.6 | 451.1 | 881.4 | 1052.5 | 570.5 | 37.8 | 11.5 | 47.2 |
| Dengue cases | 14 | 4 | 13 | 16 | 27 | 10 | 19 | 1044 | 6433 | 5042 | 4187 | 327 |

The time series analysis revealed significant changes in the monthly rainfall during the study period (2007–2012) (Figure 6a). The maximum rainfall was recorded from June to September. It also indicates that the total rainfall from 2007 to 2012 increased (positive slope 0.16). Seasonally, the slope in the given interval is much higher (positive slope 20.90) than the overall annual perspective (Figure 6b). The total monthly temperature was decreased with slope −0.003 and seasonally increased by a slope of 0.62, respectively (Figure 7a,b).

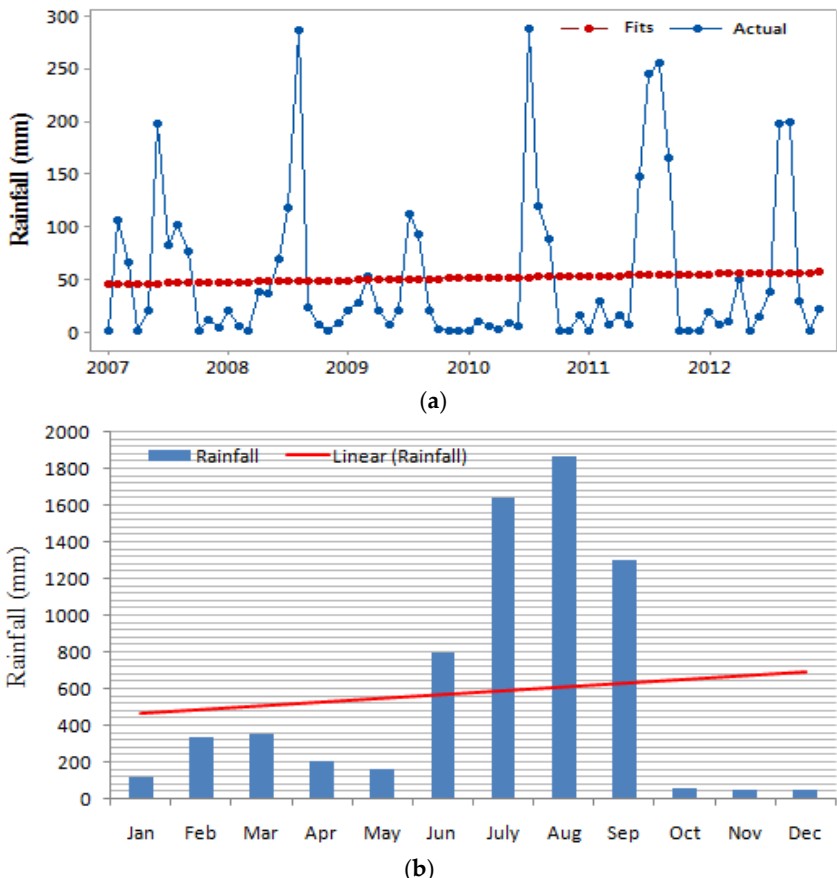

**Figure 6.** (**a**) Monthly time series of rainfall in Lahore City, 2007–2012. (**b**) Seasonal time series of rainfall in Lahore City, 2007–2012.

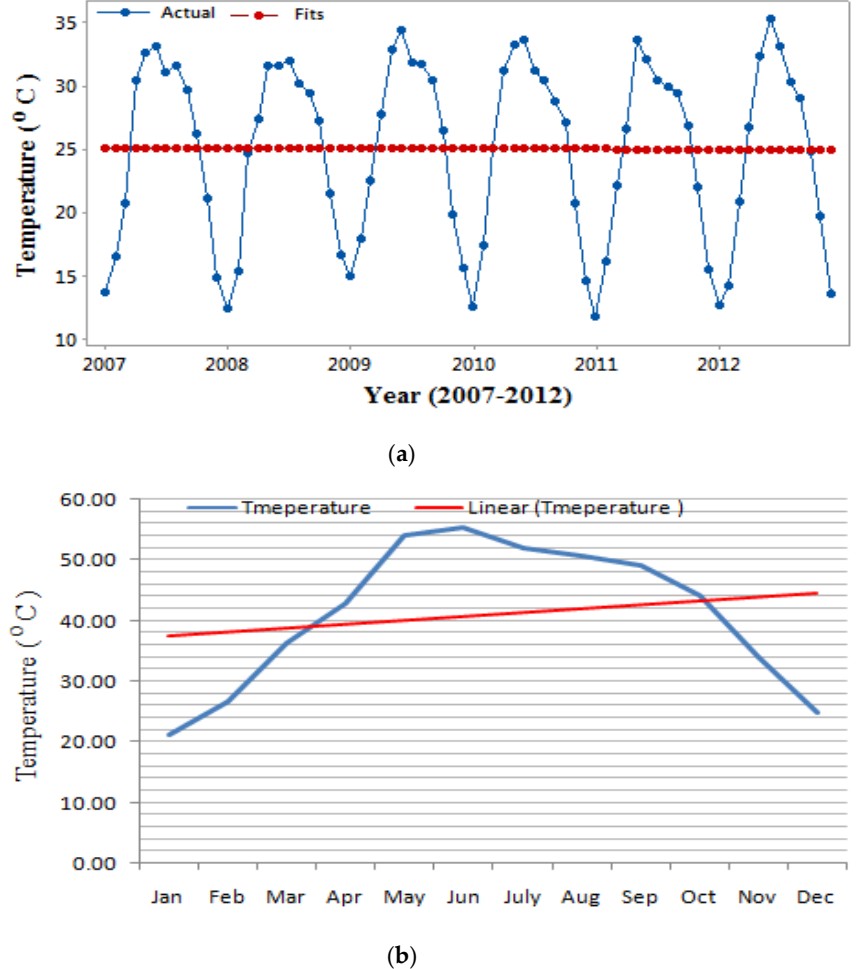

(a)

(b)

**Figure 7.** (**a**) Monthly time series of temperature in Lahore City, 2007–2012. (**b**) Seasonal time series of temperature in Lahore City, 2007–2012.

The temporal variation of monthly reported dengue cases during 2007–2012 showed an increasing trend with a slope of 8.09 (Figure 8a). A total of 16,497 DF cases were reported during the study years, while the highest number ($n$ = 11,221) of dengue cases was reported during 2011 due to extreme rainfall during that year and suitable monsoon and post-monsoon temperatures. The lowest number of cases ($n$ = 89) remained in 2009 due to considerably lower rainfall events. In every year, dengue cases showed a specific peak of incidence. Mostly, DF cases were reported during the monsoon and post-monsoon periods. During January–July, insignificant numbers of dengue cases were reported while the maximum numbers of cases were reported from August to December. DF dispersion reached the peak during September ($n$ = 6433) and progressively showed an increasing trend from January to December, as shown in Figure 8b. This particular scenario reflects that temperature and rainfall are more seasonally intensive. This condition provides ideal grounds for dengue breeding and transmission during and after monsoon times.

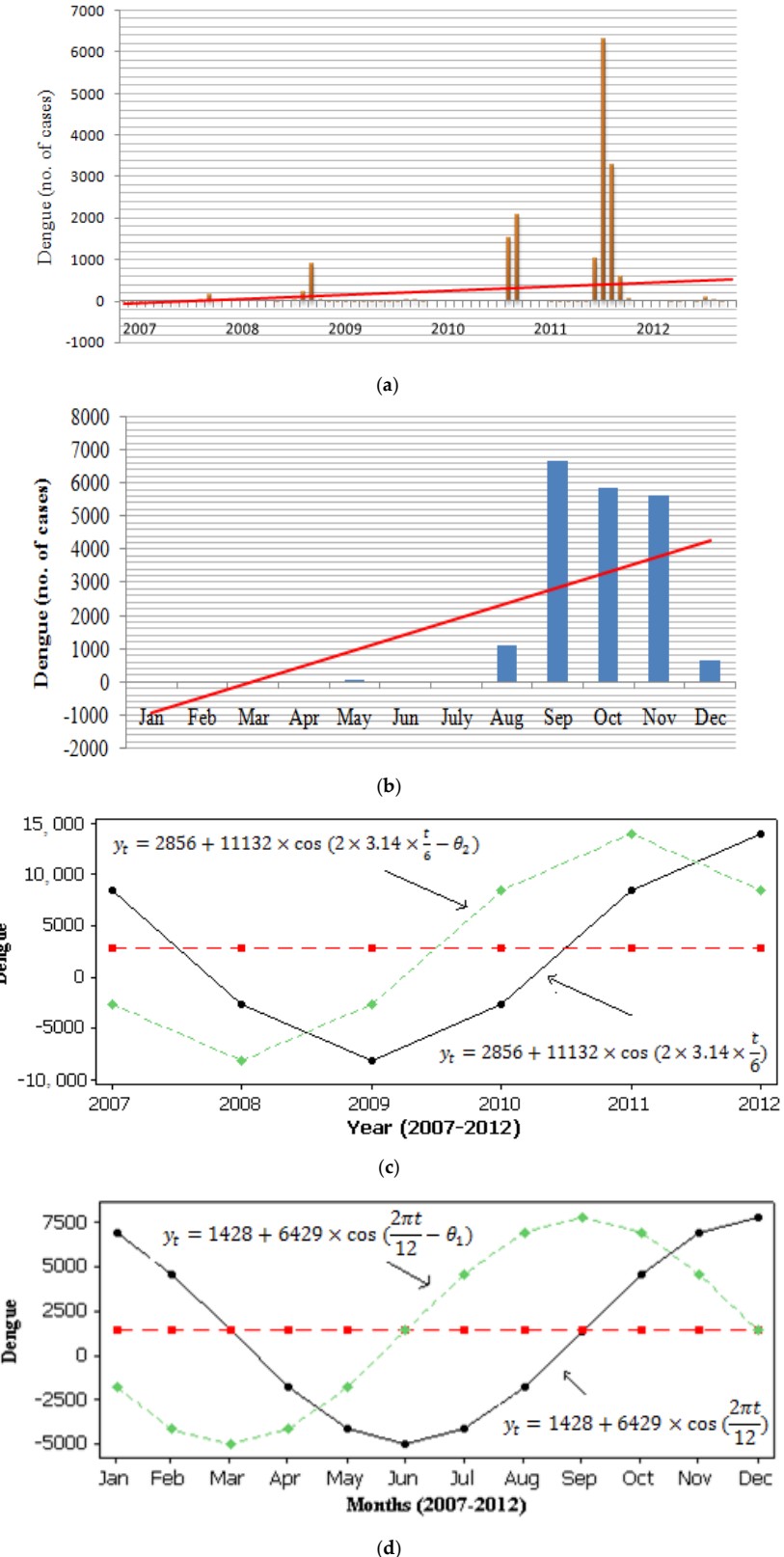

**Figure 8.** (**a**) Monthly time series of the number of reported dengue cases in Lahore City, 2007–2012. (**b**) Seasonal time series of the number of reported dengue cases in Lahore City, 2007–2012. (**c**) Transformation of cosine functions to represent an annual cycle of yearly dengue incidence for Lahore, Pakistan, for the years 2007–2012. (**d**) Transformation of cosine functions to represent an annual cycle of monthly dengue incidence for Lahore, Pakistan, for the years 2007–2012.

Figure 8c shows the cosine function for the total annual data points to display the dengue incidence from 2007 to 2012. The black line in the figure is simply a curve of six data points with t = 1, indicating the year 2007, t = 2 (2008), and so on. The overall mean annual dengue cases are indicated by a red line, and the green line indicates that the transformation of the cosine curve needs to be shifted to the right to line up well with the data. The maximum in the curve can be identified at t = 5 (the year 2011) by introducing a phase shift of $\varnothing_2 = \frac{2\pi t}{6}$ (t = 5). Choosing the amplitude C = 5566 allows to half the difference between 2011 and 2009. The result shows the highest dengue incidence in 2011. Figure 8d illustrates the foregoing procedure using the total monthly dengue incidence for 2007–2012 at Lahore. The black line in the figure indicates a simple curve of 12 data points, with t = 1 indicating January, t = 2 indicating February, and so on. The overall mean monthly incidence of 1428 cases is indicated by a red line. The black curve in the figure shows this function lifted to the level of the total monthly dengue incidence and extended so that its range can be like that of the data series. The extension was done only approximately, by choosing the amplitude C = 3214.5, to be half the difference of September and February of dengue incidence. Finally, the cosine curve needed to shift to the right for lining up well with the data. The maximum in the curve was set at t = 9 months (September) by introducing the phase shift of $\varnothing_1 = \frac{2\pi t}{12}$ (t = 9). This showed the highest dengue incidence in November. The correspondence between the curve and the data was improved by using better estimators for the amplitude and phase of the cosine function. The data appears to be sinusoidal, executing a single full cycle over the course of the 12 months.

### 3.3. Impact of Climate Variables on DF

To examine the relationship between climatic factors and DF, we also employed ordinary linear and multiple linear regression analyses on the data of 2007–2012. The monthly average temperature and monthly rainfall were used as independent variables, whereas total monthly reported dengue cases were used as the dependent variable. For a temporal evaluation, a one-way analysis of variance (ANOVA) was utilized to see if each one of the climate factors differed significantly between times.

Ordinary linear regression was performed between monthly rainfall and dengue cases. The resulting slope is 14.659, and the intercept is 2456, which indicates the dengue cases increased with a positive slope (whenever rainfall increased) (Table 3, Figure 9). The model verification was judged using the coefficient of determination $R^2$ and F-test. The F-test represents the ratio of two means (i.e., $F = \frac{MSS}{MSE}$). This indicates that the positive slope 14.7 was well considered because of the *p*-value of <0.05 and the observed F-value of 22.85 (*p* = 0.00) which is higher than the critical value of the F-test (4.001) at n − 1 degree of freedom. The coefficient of determination showed 34.2% of model significance. Lai (2018) found a coefficient of determination of 13.8% between climatic variables and dengue cases [31].

**Table 3.** Monthly ordinary linear regression analysis of dengue versus rainfall.

| Predictor | Coef | SE Coef | T | *p* | |
|---|---|---|---|---|---|
| Constant | 2455.8 | 364.1 | 6.75 | 0.00 | |
| Rainfall | 14.659 | 3.066 | 4.78 | 0.00 | |
| S = 36132.9 | | $R^2$ = 34.2% | | | |
| **Analysis of Variance:** | | | | | |
| **Source** | **DF** | **SS** | **MS** | **F** | *p* |
| Regression | 1 | 29,837,486,298 | 29,837,486,298 | | |
| Residual | 44 | 57,445,902,491 | 1,305,588,693 | 22.85 | 0.00 |
| Total | 45 | 87,283,388,789 | | | |

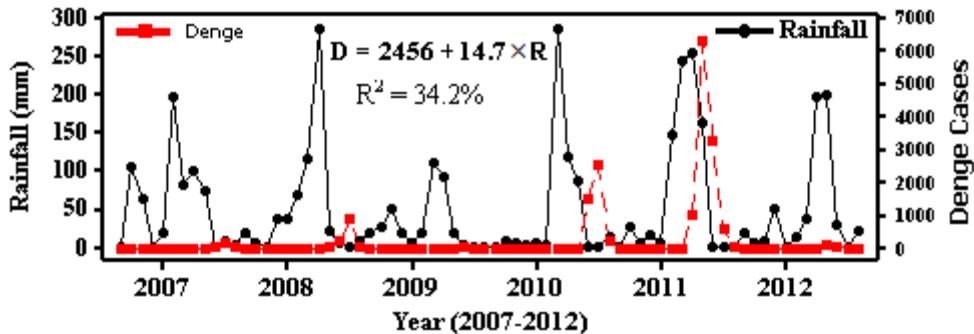

**Figure 9.** Comparison of monthly dengue cases versus rainfall with regression equation in Lahore, 2007–2012.

As in Table 4, the slope value is 371 and the intercept is −6213, which indicates that the monthly dengue incidence was also positively connected with monthly temperature (Figure 10 and Table 4). The result of the F-test was 25.79, suggesting a high significance of the model.

**Table 4.** Monthly ordinary linear regression analysis of dengue with temperature.

| Predictor | Coef | SE Coef | T | *p* |
|---|---|---|---|---|
| Constant | −6213 | 1909 | −3.26 | 0.002 |
| Temperature | 370.87 | 71.48 | 5.19 | 0.000 |
| S = 35081.0 | | R$^2$ = 38.0% | | |

**Analysis of Variance:**

| Source | DF | SS | MS | F | *p* |
|---|---|---|---|---|---|
| Regression | 1 | 33,133,717,525 | 33,133,717,525 | | |
| Residual | 44 | 54,149,671,264 | 1,230,674,347 | 25.79 | 0.013 |
| Total | 45 | 87,283,388,789 | | | |

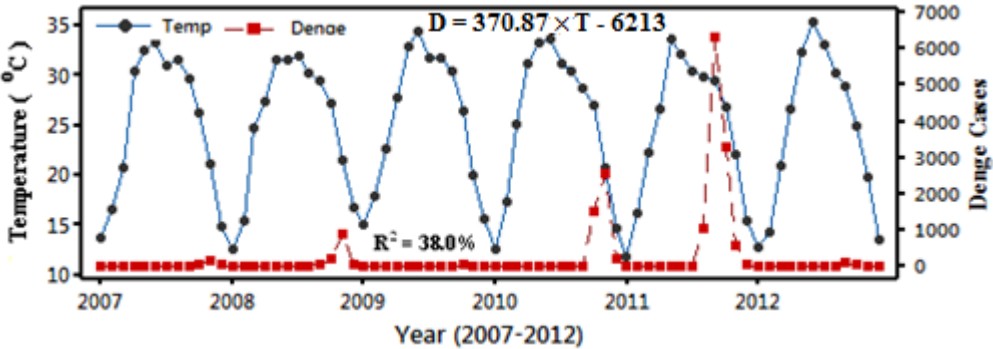

**Figure 10.** Comparison of monthly dengue versus monthly temperature with regression equation in Lahore, 2007–2012.

The observed model accuracy was 38.0% at a 95% confidence interval. The respective *p* value of <0.05 and F-test value of 25.79 also suggested the significance of the model. The comparative results of the multiple regression analysis showed a positive relationship between monthly dengue cases and climatic factors (temperature and rainfall) (Table 5, Figure 11). The coefficient of determination for this model is 44.6% which is significant at a 95% confidence interval with a *p*-value of <0.05. Karim et al. (2012) applied a significant multiple linear regression model between dengue cases and climatic variables. They found a coefficient of determination of 26.4% for the first model for the total data, 50.9% for the second model for one lag month, and 61.0% for third model with an average of two lag months [30].

**Table 5.** Multiple regression analysis of monthly dengue versus mean monthly temperature and total monthly rainfall in Lahore.

| Dengue | Coef. | SE Coef | T | *p* |
|---|---|---|---|---|
| Constant | −4967 | 2763 | −1.80 | 0.079 |
| Temperature | 300.1 | 111.7 | 2.69 | 0.010 |
| Rainfall | 6.744 | 4.240 | 1.59 | 0.019 * |
| S = 33,857.2 | | $R^2$ = 44.6% | | |
| **Analysis of Variance** | | | | |

| Source | SS | df | MS | F | *p* |
|---|---|---|---|---|---|
| Model | 38,695,350,650 | 2 | 19,347,675,325 | | |
| Residual | 48,145,018,459 | 69 | 1,146,309,963 | 16.88 | 0.000 |
| Total | 86,840,369,109 | 71 | | | |

\* means *p*-value not significant.

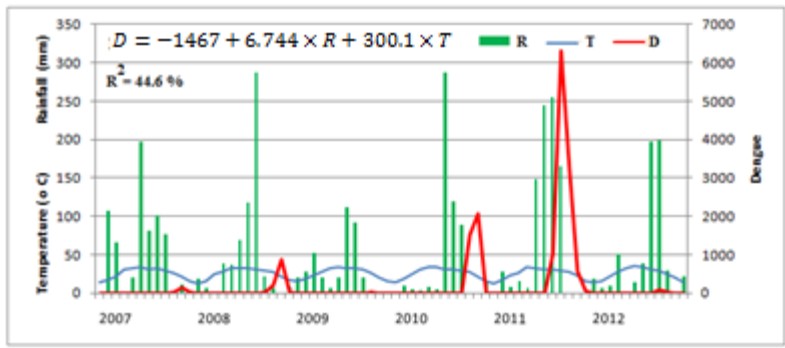

**Figure 11.** The plot of total monthly dengue (D) versus climatic factors (i.e., monthly temperature (T) and rainfall (R)) with multiple regression equation in Lahore, 2007–2012.

## 4. Discussion

Dengue has been evolving due to many factors, including ineffective disease and vector observations; insufficient public health infrastructures or organizations; an increase in population; unplanned and uncontrolled urbanization; and changing climatic conditions [41]. However, changing climatic conditions are considered as the major cause of its spread and emergence because dengue is largely dispersed in hot and humid areas of the world [42–45]. In numerous studies, the close relationship between dengue and climatic factors was detected, especially in terms of their seasonal arrangements [43,46–50].

In the present study, we have utilized statistical approaches to find out the variability of climatic factors and their effects on dengue incidence in Lahore district. Previously, people have successfully used trend lines as a tool of prime importance to study the economic development, hydrological planning, and climate change of a country [39]. Therefore, we employed this technique to find the trend of temperature and rainfall in Lahore. We noticed that the trend line of climatic factors increased during the 43 years studied. This increasing trend line indicates that the climate is significantly changing in Lahore [27]. We observed that the temperature and rainfall were suitable for DF breeding and dispersal from monsoon to post-monsoon times. The post-monsoon time was especially significant because of a high incidence peak of DF during this particular phase every year (rainfall occurred earlier than dengue incidence). This seasonal suitability might have led DF to its climax during the highest incidence year, i.e., 2011. These results are in correspondence with a study in Mexico, which discovered that when the rainfall quantity became suitable, dengue incidence accelerated. The authors noticed that when it rose up to 550 mm, dengue was increased [51].

These changing climatic conditions have made Lahore district attractive for DF emergence. The results of our study are in line with other studies that also illustrate the positive influence of temperature and rainfall on reported dengue cases. As discussed above, dengue seems to have made a comeback in Lahore after 2011's massive epidemic. Climatic suitability might have played a greater role in its reemergence, but ever-decreasing governmental and public attention after 2011's epidemic may also have helped it. Our statistical results further confirm that the incidence of dengue is likely to increase in Lahore due to a positive change in these climatic factors. These findings are concurrent with previous studies, where the frequency and time with respect to temperatures and temperate rainfall seem to play an important role in sustainable dengue transmission [52]. Moreover, it is reported that dengue cases are more abundant during the rainy season because of the increasing formation of reproduction locations for mosquitos [41].

Pakistan has a subtropical temperate climate ruled by the summer monsoon, which creates hot and humid conditions (especially post-monsoon). These post-monsoon conditions are considered very favorable for the dengue mosquito's (*Aedes aegypti*) replication and maturation. These favorable conditions are further complicated by a poor infrastructure, erratic water supply which forced the residents to store water in containers for domestic use, low educational status of the residents or high level of illiteracy, poor sanitation, and increase in population growth [53,54]. Although these factors contribute significantly to the high mosquito density, the climate is the major factor that primarily controls the DF transmission.

Risk analysis of this kind is considered a fundamental step for other advanced analyses, e.g., an early warning system (EWS). It can help the DF controlling plans devised for the study area by the Government's public health departments. We encourage them to develop an EWS for an informed decision making against DF reemergence [55,56].

## 5. Conclusions

This study is comprised of the monthly as well as the annual variation of climatic factors (1970 to 2012) and their role in triggering the dengue disease from 2007 to 2012. The resultant significant highlights of various analyses performed are presented as follows:

i. The annual and monthly mean temperatures increased to 25.07 °C with a standard deviation of 7.074. The rainfall also increased annually but decreased monthly from 1970–2012.

ii. During 2007–2012, the monthly total rainfall increased, while means of the monthly temperature decreased and dengue cases increased, respectively.

iii. Events of maximum total rainfall were recorded (2007–2012) during 2008, 2010, 2011, and the maximum dengue cases occurred during the years 2010 and 2011, as shown in Figure 8a,b. The climatic factors, especially temperature and rainfall, affect several regions of the world, i.e., the environmental as well as the health sector. It is also known that rainfall provides a medium for the aquatic stages of the dengue mosquito's life cycle, while, in addition, the temperature provides optimum conditions for mosquitos to breed and multiply. The significance of analysis revealed that the incidence of the high peaks of dengue case was after the monsoon season in every year, and the climatic event, i.e., rainfall, occurred earlier than dengue incidence, as shown in Figure 8c.

iv. The results of ordinary linear and multiple regression analyses reveal a good relationship between dengue cases and climate variables; if rainfall increased then dengue cases also increased with a model accuracy of 34.2%. Dengue incidence also increased when the mean monthly temperature increased. Moreover, multiple regression also indicated a positive relationship between climatic factors and dengue cases, i.e., 44.6%. This model showed a significant increase in dengue when the temperature and rainfall increased.

v. If we further increase the number of climatic factors, i.e., humidity, sunshine, and emissions (environmental factors), then the model will be improved.

Finally, our assumptions conclude that climatic factors, i.e., both rainfall and temperature, enhanced the dengue incidence in densely populated areas like Lahore city. We here recommend to the researchers and policymakers to initiate auxiliary and future studies to develop the continuous and long-term sustainability of prediction accuracy, which will help to maintain the performance of a forecasting model. Furthermore, if more advanced research is initiated for turning the prediction model into a user-friendly or nontechnical equipped mechanism, such efforts can make it understandable for common users and will encourage them to extensively cope with dengue's threat.

**Author Contributions:** Conceptualization, S.A.A.N.; Data curation, S.J.H.K.; Formal analysis, S.A.A.N. and B.J.; Investigation, S.A.A.N.; Methodology, S.A.A.N. and B.J.; Project administration, S.A.A.N.; Resources, S.S.; Software, B.J.; Supervision, S.S. and S.J.H.K.; Validation, S.J.H.K.; Writing–original draft, S.A.A.N.; Writing–review & editing, L.A.W., M.N.-u.m. and N.A.

**Funding:** This research received no external funding.

**Acknowledgments:** We thank "Directorate General Health Services, Punjab" (DGHSP) and "Pakistan Meteorological Department, Lahore" (PMD) for providing us with the dengue and climatic datasets.

**Conflicts of Interest:** The authors declare no conflict of interest.

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
