# Peer review of "Changing Climatic Factors Favor Dengue Transmission in Lahore, Pakistan"

_environments, doi:10.3390/environments6060071_

Round 1

Reviewer 1 Report

A study on the  the climatic impacts of temperature and  rainfall) on Dengue Fever disease in district Lahore (Pakistan) is presented.

The authors consider time series of temperature and rainfall in Lahore from 1970 to 2012 by considering  monthly time series from 2007 to 2012. They apply a linear regression analysis to monthly data e a multiple regression analysis to annual data to analyze changement of the dague fever incidence..

In the intruduction section, a more rigorous investigation on existing climatic impacts methods, such as comparison of previous approaches in terms of pros and cons, should be given. In addition, the authors should highlight the reasons for using simple statistical approaches to study the monthly and annual time series, instead of more sophisticated methods as ARIMA or nonlinear artificial neural networks and support machine methods. 

Which comparative tests did the authors carry out with respect to the tests performed applying linar and multiply regression equations to justify their conclusions in section 5?

Author Response

Title:  Changing climatic factors favor Dengue transmission in Lahore, Pakistan

Responses with reference to the suggestions:

Reviewer: 1

We thank the reviewer for keenly reading our manuscript. It has helped us in improving our article. We have given responses to your suggestions and also edited the language. In the revised manuscript, red colored words are changes in response to the reviewer’s comments whereas improvement of English language is presented in green color.

1.      In the introduction section, a more rigorous investigation on existing climatic impacts methods, such as comparison of previous approaches in terms of pros and cons, should be given. In addition, the authors should highlight the reasons for using simple statistical approaches to study the monthly and annual time series, instead of more sophisticated methods as ARIMA or nonlinear artificial neural networks and support machine methods. 

Answer:

We appreciate your comment and as per your suggestion we have highlighted the importance of using simple statistical approaches as mentioned form line 72 to 84.

Multiple Linear Regression (MLR) is used for solving such problems. Because the prediction is made on numeric continuous variables. Along with this, the problem also includes time series forecasting. A large number of successful applications has shown that MLR algorithms can be very useful tools for time-series modeling and forecasting. This research is a part of a larger research project which compares the impact of climate change on Dengue incidence over Pakistan and the other approaches i.e.  ARIMA or nonlinear modeling and artificial  neural networks, are also important. Such comparisons will appear in other communications.

2.      Which comparative tests did the authors carry out with respect to the tests performed applying linear and multiple regression equations to justify their conclusions in section 5?

Answer:

Many researchers (Karim, et al., 2012, Colón-González, et al., 2011,etc.) justify the MLR models on the bases of F-test,  R2, and p-value. We have also applied a similar test to check the performance of models in this study.

Reviewer 2 Report

Overall, the paper presents a study of a significant public health issue and it is well written. Some statistical analysis needs to be improved before publication.

Section 3.1: Time-series mean values should be illustrated in addition to standard deviation values.

Figures 6 (a, b) and 7(a, b): It seemed that the in all four figures, climate variables (rainfall or temperature) were regressed against time, which does not make sense to me. I suggest removing all the equations from the figures and only showing the time-series plots and the mean values as a constant line.

Figures 9b, 10b and 11: In all the three regression analyses, n=6, which is too small to generate reliable results. I suggest the authors consult with a statistician and at least discuss the pitfall of small n in a regression analysis and how it may bias the result of this study.

Line 306: "It is evident from results that during 43 years, trend line of climatic factors has increased" - again, it is hard for me to see from Section 3.1 since only standard deviations were presented in the figures.

Line 362: "94.5%" - where in the paper this result was presented?

Author Response

Title:  Changing climatic factors favor Dengue transmission in Lahore, Pakistan

Responses with reference to the suggestions:

Reviewer: 2

We thank the reviewer for keenly reading our manscript. It has helped us in improving our article. We have given responses to your suggestions. In the revised manuscript, red colored words are changes in response to the reviewer’s comments whereas improvement of English language is presented in green color.

Top of Form

1.      Section 3.1: Time-series mean values should be illustrated in addition to standard deviation values.

Answer: The comment has been accommodated accordingly as the reviewer suggested.

2.      Figures 6 (a, b) and 7(a, b): It seemed that the in all four figures, climate variables (rainfall or temperature) were regressed against time, which does not make sense to me. I suggest removing all the equations from the figures and only showing the time-series plots and the mean values as a constant line.

Answer: The comment accommodated accordingly.

3.      Figures 9b, 10b and 11: In all the three regression analyses, n=6, which is too small to generate reliable results. I suggest the authors consult with a statistician and at least discuss the pitfall of small n in regression analysis and how it may bias the result of this study.

Answer: Thanks for highlighting this point, we have removed annual data comparison from series, it is so small. It should be more than 30 at least.

4.      Line 306: "It is evident from results that during 43 years, the trend line of climatic factors has increased" - again, it is hard for me to see from Section 3.1 since only standard deviations were presented in the figures.

Answer: The correction has been done accordingly as per reviewer’s suggestion.

5.      Line 362: "94.5%" - wherein the paper this result was presented?

Answer: Thanks for pointing it out. The result is removed from the text.

Reviewer 3 Report

The manuscript addresses an issue that is continuously being investigated, which is how dengue is affected by climate. Comments follow.

-      Ln. 49: references are required.

-      Ln. 190: how do you know at this point this is the case?

-      What is t he figure on page 8, line 195 that is not labeled?

-      Ln. 236: please correct slop, it should be “slope”.

-      Ln. 241: 34.2% is not a good R2. Can you please add references that show in this type of research this is considered a good value.

-      Ln. 248: this regression is only using 6 observations (one for each year). This is not enough. The sample is too small. This is perhaps what is producing an R2of 87.9%.

-      Ln. 266: with temperature? Please clarify.

-      Figures 10, 11, and 12 with their corresponding tables, are they necessary? 

-      Editing required.

Author Response

Title:  Changing climatic factors favor Dengue transmission in Lahore, Pakistan

Responses with reference to the suggestions:

Reviewer 3

We thank the reviewer for keenly reading our manscript. It has helped us in improving our article. We have given responses to your suggestions. In the revised manuscript, red colored words are changes in response to the reviewer’s comments whereas improvement of English language is presented in green color.

1.      Ln. 49: references are required.

Answer: The references are accommodated accordingly.

2.       Ln. 190: how do you know at this point this is the case?

Answer: Due to the highest number (n=11221) of dengue cases during 2011 and extreme rainfall during these years. (please see from line 200 to 202)

3.      What is the figure on page 8, line 195 that is not labeled?

Answer: Labeled Figure 8 (b) now in line 207

4.       Ln. 236: please correct slop, it should be “slope”.

Answer: Corrected slop as the slope

5.       Ln. 241: 34.2% is not a good R2. Can you please add references that show in this type of research this is considered a good value?

Answer: References for this point are Lai, Y. H. (2018) and Karim et al., (2012) and explained in literature also.

6.      Ln. 248: this regression is only using 6 observations (one for each year). This is not enough. The sample is too small. This is perhaps what is producing an R2of 87.9%.

Answer: Thanks for highlighting this point, we have removed annual data comparison from series, it is so small. It should be more than 30 at least.

7.       Ln. 266: with temperature? Please clarify.

Answer: Monthly linear regression of dengue cases with temperature.

8.      Figures 10, 11, and 12 with their corresponding tables, are they necessary? 

Answer: Arranged the figures and tables accordingly

9.      Editing required.

Answer: We appreciate your comment and we have edited the whole manuscript where required.

Round 2

Reviewer 1 Report

In this new version of their paper the authors take into account all my suggestions. I consider this paper publishable in the present form.

Author Response

We thank the reviewer for reviewing this manuscript and valuable time. We appreciate your positive and encouraging comments.

Reviewer 3 Report

The authors have tried to address the reviewer comments. The manuscript is clearer now. Comments follow.

-       The methods used are well known. So what is the contribution exactly? Others have shown that weather change impacts the appearance/ resurgence of dengue. Is it only a case study of the findings in Lahore? This should be made clear.

-       Ln. 237: the authors should revise the language. Perhaps “developed statistical model” is not the right term. The authors applied an existing model. “Developed” sounds like it is a new model introduced by them, which is not the case.

-       Lns. 267-269: what models are these?

Author Response

We thank the reviewer for reviewing this manuscript and valuable suggestions. Here are the responses to reviewer’s comments. In the revised manuscript, red colored words are changes in response to the reviewer’s comments.

1.      The methods used are well known. So what is the contribution exactly? Others have shown that weather change impacts the appearance/ resurgence of dengue. Is it only a case study of the findings in Lahore? This should be made clear.

Answer: Yes, you are right that these methods are well known, however, our contribution is the use of existing methods and interpretation of the data. We have focused on area of Lahore particularly, where Dengue has remained an epidemic. We have used climate data of 43 years and utilized it to show the climate change in Lahore over the years. 

We have various findings related to DF-climate relation in Lahore, one of those is increasing trend of DF in post monsoon times. This study focuses only on Lahore city only where dengue is on the rise and this study could help in planning of Early warning system for Dengue incidence in the city.  

2.       Ln. 237: the authors should revise the language. Perhaps “developed statistical model” is not the right term. The authors applied an existing model. “Developed” sounds like it is a new model introduced by them, which is not the case.

Answer: The sentence has been revised.

3.       Lns. 267-269: what models are these?

Answer: These models are multiple linear models but the difference is the lag time of one month and two months from one even to another related event. 

This manuscript is a resubmission of an earlier submission. The following is a list of the peer review reports and author responses from that submission.

Round 1

Reviewer 1 Report

Modelling monthly dengue counts using a linear regression model [see Fig 8c] assumes that the monthly dengue counts are mutually independent, which I do not feel is a reasonable assumption.

There are numerous grammatical and spelling errors throughout. Also, inconsistent notation is used for the same quantities (for example, Yt, Y(t), Y_t, y). Also, predictions should be denoted Y hat, not Y(t).

Some terms used in the tables are not defined (for example, in Table 1(a), SE Mean is not defined.

Reviewer 2 Report

It is difficult to recommend the paper in its current form. This text will require significant editing to address language, but there are more general difficulties with the statistics. Namely: OLS models are used and tested inappropriately, and cyclicity is only briefly acknowledged with a set of cosine fits that seem ad hoc and unrelated to the rest of the analyses. Some claims are made that are not apparently supported by the data. Ideally, we would see something like a Poisson model with cases as a function of lagged temperature and precipitation.

Miscellaneous:

Line 98 and throughout: "data" are typically plural.

Lines 108-119: highly redundant language and probably unnecessary. It should suffice to mention OLS with dengue incidence as a function of temperature and rainfall. Instead, make some mention of software used for calculations.

Lines 130-131: it isn't clear where this claim is demonstrated. If Figure 7 is to be believed, we're looking at decreasing temperatures? But the OLS model is definitely inappropriate here for the monthly data with its heavy cyclicity.

Line 165: minimum annual dengue cases are not listed, although it's clearly 0.

Lines 195-218: it is clear that this is an attempt to handle cyclicity, but this should be a term in the OLS model rather than a separate fit by hand. Unclear what these terms contribute at all.

Line 190: it does not appear to be true that 2011 had large variability in precipitation in Fig 4, so it's unclear where this claim comes from.

Figure 8: the OLS model is not appropriate here. While it does capture the increasing trend over time, the model is zero-inflated and this should probably be done on an annual level.

Line 247: should be p-value instead of the test statistic.

Line 249: No confidence interval is given here.

Figure 10: "denge" rather than "dengue"

Lines 294-296: redundant

Line 316: Unclear what's being cited. Is this [42]?

Misc. errors in lines 38, 40, 48, 62, 69, 80, 100, 176, 229, 305, 310.

Reviewer 3 Report

General comments:

1.      Generally, not only temperature and rainfall have impact on DF incidence, humidity is also an important factor, so directly using these two climatic factors to do analysis, rather than selecting influential factor first, is not convincing.

2.      Many previous studied have been conducted on investigating the relationship between weather factors and dengue incidence in different areas worldwide. And most researches indicate they show non-linear relationship(see some listed references). In this study, the authors used ordinary linear regression analysis and multiple regression analysis to detect the relationship is not reasonable. Moreover, there are three major issues:

1) Climate change affect dengue incidence is mainly embodied by its seasonal variation. Long-term effect could be detected using yearly data, however in this study only 6 years data (2007-2012) were used to build the relationship equation. So it is not meaningful to study the annual correlation.

2) These results were verified by Coefficient of determination (R2), the finding “ordinary linear regression of rainfall versus dengue showed monthly R2=34.2% … whereas temperature versus dengue presented monthly R2=38.0% …” cannot ensure the equation is statistically significant.

   3) The study only used 6-year data to develop the model, but there lack a verification of the developed model using date in different years.

Based on the above reasons, I do not think this paper is proper to be published in the journal.

Specific comments

1)      Some figures are plotted non-standard. For example, only one caption is enough for both Figure (6) , Figure (7) , Figure (8) and Figure (9) .

2)      Some language mistakes, such as line 175, line 277, “and climatic event i.e. rainfall” in line 293.
